# Comparison of In Vitro Methods for Assaying the Antibacterial Activity of a Mix of Natural Essential Oils Against Zoonotic Bacteria

**DOI:** 10.3390/microorganisms13051125

**Published:** 2025-05-14

**Authors:** Karine Fayolle, Claire Girard, Pauline Lasfargues, Sahar Koteich, Sylvain Kerros

**Affiliations:** 1INRAE, VetAgro Sup Campus Agronomique de Lempdes, UMR F, Université Clermont Auvergne, 15000 Aurillac, France; 2Microlab—Phytosynthese, 63200 Mozac, France; claire.girard@phytosynthese.com (C.G.); lasfarguespauline@gmail.com (P.L.); saharkoteich2000@gmail.com (S.K.); 3Phytosynthèse, 63200 Mozac, France; sylvain.kerros@phytosynthese.com

**Keywords:** essential oils, plant extract, antibacterial, MIC—microdilution method

## Abstract

With the increasing occurrence of bacterial resistance, it is now essential to look for new alternatives to protect the curative utilization of antibiotics within the One Health concept. Here, we adapt and optimize a broth microdilution method and compare it against the broth macrodilution method for evaluating the antibacterial activity of a complex essential oils mix (EO mix) against four livestock pathogens: *Escherichia coli*, *Bacillus cereus*, *Pseudomonas aeruginosa*, and *Staphylococcus aureus.* Microdilution method performance (final volume well: 300 µL; inoculum: 1.0 × 10^6^ CFU/mL) was evaluated following CLSI recommendations, by comparing the MIC of each of the four strains with the MICs obtained with the macrodilution method (final volume tube: 2 mL; inoculum 1.0 × 10^6^ CFU/mL). Microdilution analysis was performed with an automated plate reader (Bioscreen C), and three bacterial growth parameters (OD max, lag phase, and growth rate) were calculated (DMFit curve-fitting software (v2.1; courtesy of the Institute of Food Research, Norwich, UK)). EO mix MICs were determined for *E. coli*, *S. aureus*, and *B. cereus*. Our results emphasize the importance of ensuring the accuracy of MIC results by performing three technical and three biological replicates, and combining OD max, lag phase, and growth rate to assess the impact of an EO mix at sub-MIC levels.

## 1. Introduction

Antibiotics have revolutionized modern medicine, enabling treatment for life-threatening infections. However, with the increasing occurrence of bacterial resistance against current antibiotics, it has now become essential to look for newer substances or even alternative methods to compensate for decreases in antibiotic power [1]. In fact, the European Union Summary Report on Antimicrobial Resistance reported worrying proportions of multidrug-resistant *Escherichia coli* isolated from broilers (37.7%), turkeys (47.6%), pigs (28.8%), and veal calves (18.8%) in 2020–2021 [1]. The use of essential oils (EOs) or other natural plant extracts emerges as a sustainable alternative solution to limit the use of antibiotics.

EOs, especially those that mainly contain phenolic compounds, have natural antibacterial, antimicrobial, antiviral, and antifungal activities that help protect the host plant [2,3]. However, little is known about the potential use of EOs in livestock. Findings in the relevant literature concern the effects of medicinal plant species on livestock disease but do not go deeper into the isolated effects of their individual bioactive compounds. There are also only limited data on EOs [4]. However, it is important to study the efficacy of EOs on livestock pathogens, as EOs have been proposed for use as an alternative treatment to antibiotics in order slow the emergence of resistant bacteria [5,6]. Livestock animals are vulnerable to intestinal infections, especially during the weaning period, but the available antibiotic treatments are losing efficacy due to rising antibiotic resistance.

*Escherichia coli* (*E. coli*), *Staphylococcus aureus* (*S. aureus*), *Bacillus aureus* (*B. cereus*), and *Pseudomonas aeruginosa* (*P. aeruginosa*) are commonly used as model organisms due to their diversity in cell morphologies and respiratory types, as well as their involvement in foodborne and/or zoonotic diseases. Several recent in vitro studies have demonstrated the antibacterial action of EOs on many major foodborne pathogenic bacteria that are also involved in diseases of livestock, e.g., oregano oil, thyme oil, and clove or lemongrass oils against *E. coli* [7]; clove oil against *S. aureus* [8]; citrus oil against *Salmonella* [9]; and oregano oil, thyme oil, clove oil, and cinnamon oil against *Listeria* spp. [10]. *P. aeruginosa* is among the most abundant infectious microorganisms for animals and humans [4], causing respiratory infections such as pneumonia and enteric infections. Chickens are particularly vulnerable to intestinal infections caused by this zoonotic pathogen, and chicken carcasses and retail poultry products are a major vector of *P. aeruginosa* transmission to humans, especially after processing in abattoirs. Antibiotic resistance has made *P. aeruginosa* infection extremely difficult to treat [11]. *B. cereus* was identified as a sporulated model organism, as it is a known foodborne pathogen that produces enterotoxigenic toxins and has zoonotic potential due to its ubiquitous presence in soils [12]. *S. aureus* and *E. coli* cause diseases in monogastric animals, ranging from respiratory or intestinal infections to severe systemic infections [13], and have various implications in the processing and production of meat products [14]. Moreover, the formation of biofilms by *S. aureus* makes infections more challenging to treat [14].

Natural plant extracts, including EOs and various ethanolic or methanolic dried extracts, are also widely used as nutritional additives to enhance growth parameters such as feed conversion, average daily gain, carcass quality, or milk yield [15]; to decrease oxidative stress and inflammation; and to strengthen immunity [4]. In addition, all these mentioned effects work concomitantly with the prevention of enteropathies in farm animals [7]. The antibacterial and especially bacteriostatic action of phytogenic compounds have been demonstrated in vitro [16]. In vivo, their antimicrobial effect works by modifying the microflora ecosystem. Some EO compounds, such as cinnamaldehyde, thymol, and carvacrol [17,18], may selectively influence gut microflora. Cinnamaldehyde, for instance, was shown in vitro to be strongly inhibitory for coliform bacteria and *E. coli* [19].

Although many studies have explored the antibacterial effects of EOs [20,21,22,23], there is no standardized reference method and no universally accepted indicators for repeatability and reproducibility. Tools for rapid and high-throughput screening of antibacterial natural plant extracts provide the means to quickly target livestock bacterial pathogens.

The Clinical Laboratory Standards Institute (CLSI) [24,25] has standardized the agar dilution method to determine the quantitative bactericidal activity of antibiotics. The CLSI has also issued recommended broth dilution methods for the determination of minimal inhibitory concentrations (MIC) [24] that use different principles to assess microbial growth or inhibition. MIC is defined as the lowest concentration of an antimicrobial agent that prevents visible growth of a microorganism in an agar or broth dilution susceptibility test [20]. Conventional methods such as the disc diffusion method and dilution tube method can prove time consuming and require large quantities of test materials, along with other problems. Furthermore, plant extracts such as EOs are barely miscible in aqueous solution, since sedimentation quickly becomes a problem, and the interpretation of test results remains subjective and sometimes complicated by the colored plants extracts.

A number of other antibacterial assays only give an approximation of bacterial growth as they are based on visual observation of turbidity [8], which give subjective results that may not be reproducible between different laboratories. Microdilution-method MIC determination for EOs and plant extracts in general is a complex process. Many authors have worked with the CLSI recommendations and developed modified methods, with most of the effort focused on improving technical parameters. Other authors have focused on calculating parameters based on the OD, such as growth rate, compared to controls [26]. Vanegas et al. (2021) [27] reported that there are currently various in vitro approaches for evaluating the antimicrobial activity (MIC) of natural compounds, but despite the number of studies published, results fail to converge due to the fact that studies have used different methods that are not internationally standardized. Here, we did not propose a new method per se, but rather a methodology for standardizing an antimicrobial screening method in each laboratory that could contribute to ultimately developing an international consensus method. To overcome these drawbacks, two methods were compared, and an assay with modified dilution and agitation parameters to yield a ‘true’ MIC value was proposed, along with more information on target-strain growth parameters. By challenging the substance tests and bacterial targets with standardized parameters of the MIC method, our protocol can provide reproducible and meaningful results. In this study, a detailed description of the modified method of MIC determination that is sensitive, rapid, robust, reliable, and automatable is presented, and it can be successfully used to assess the antibacterial properties of natural plant extracts/EOs. The aims of this study were: (i) to evaluate the antibacterial activity of a complex mix of EOs (called EO mix) as an MIC based on by macrodilution and microdilution methods for gram-positive (*S. aureus*, *B. cereus)* and gram-negative (*E. coli*, *P. aeruginosa)* bacteria; and (ii) to discuss and suggest recommendations for the use of the most appropriate or complementary methods and indicators for MIC determination by the microdilution method.

## 2. Materials and Methods

### 2.1. Composition of the EO Mix

The EO mix was provided by Phytosynthese (Mozac, France). It is a blend of natural EOs with antimicrobial activity against gram-negative pathogens. Its composition was analyzed in triplicate using a Thermo Fisher Trace 1300 gas chromatography (GC) system (Thermo Fischer Scientific, Waltham, MA, USA) coupled to a Thermo Fisher ISQ LT mass spectrometry (MS) system (Thermo Fischer Scientific, Waltham, MA, USA). The carrier gas was helium used at a flowrate of 1.5 mL/min. Column temperature was initially 40 °C then gradually ramped up at 4.7 °C/min to 250 °C. Diluted 0.5 µL samples were injected. The components of the EO mix were identified by comparing their mass spectra against the NIST 5 library of mass spectra (National Institut of Standards and Technology, Gaithersburg, MD, USA).

### 2.2. Bacterial Strains

The bacteria were obtained from the VetAgro Sup (France) research school collection. Four strains of gram-negative bacteria, *Escherichia coli* (CIP 59.8T) and *Pseudomonas aeruginosa* (ATCC 27853), and two strains of gram-positive bacteria, *Bacillus cereus* (wild strain) and *Staphylococcus aureus* (ATCC 25923), were used. Cultures of the bacteria were maintained in aliquots of MHB (Mueller–Hinton Broth, Biokar, Beauvais, France) with glycerol (10%) at −80 °C throughout the study and used as stock cultures. Bacteria strains were revitalized in MHB (30 °C, 24H).

### 2.3. MIC by the Dilution Tube Method

All bacterial strains were tested for MIC determination using a broth dilution method following NCCLS protocols [24,25,28]. Briefly, triplicate tubes containing 5 mL of MHB medium were added with EO mix diluted as follows: progressive dilutions of EO mix were prepared 2:1 in Mueller–Hinton (MH) broth from 1⁄2 (*v*/*v*) to 1/16,384. An overnight culture of strain grown in MHB broth at 37 °C was added to target a final concentration of 10^6^ CFU/mL. Tubes were tightly capped, incubated at 37 °C, and mixed by inversion. After 24 h of incubation, the tubes were examined for visual inhibition of growth and a lack of color change of the MHB medium. MIC was defined as the lowest concentration of EOs showing inhibition of visible growth (turbidity) and a lack of color change in the MHB medium compared with the color of the controls.

### 2.4. MIC by the Automated Turbidimeter Measurements

The MICs of the EO mix against *E. coli*, *B. cereus*, *P. aeruginosa*, and *S. aureus* strains were determined using the broth microdilution method according to Kwieciński et al. (2009) [29]. All bacterial strains were incubated at 37 °C in MH broth until the stationary growth phase was reached. A diluted bacterial suspension was added to a 100-well microtiter plate (Honeycomb microplate, Oy Growth Curves AB Ltd., Helsinki, Finland) at a final concentration of 1.0 × 10^6^ CFU/mL based on spectrophotometric absorbance measured at 600 nm (Shimadzu, 1280 UV-VIS, Kyoto, Japan). Serial twofold dilutions of EO mix were prepared and added to each well to obtain a final concentration range from 5.0 to 0.005 μg/mL. All wells contained from 1% to 5% DMSO (dimethylsulfoxide) (*v*/*v*) to enhance the solubility of the EO mix. In addition, there were solvent controls (test bacteria and MHB containing from 1% to 5% DMSO), bacterial controls (test bacteria and MHB), blank controls (MHB containing from 1% to 5% DMSO and corresponding concentrations of EO mix), blank solvent controls (MHB containing from 1% to 5% DMSO), and blank medium controls (MHB). All plates were incubated at 37 °C for 24 h, and growth was evaluated by an automated turbidimeter (Bioscreen C, Labsystems, Helsinki, Finland). Optical density (OD) measurement at the 420–580 nm wavelength band was performed every 15 min after shaking for 20 s. The wideband filter (420–580 nm) is less sensitive to color changes than the other filters so the results are not affected so much by the change of color of the growth medium. MIC was defined as the lowest concentration of each antimicrobial agent for which there was no detectable growth for 24 h in liquid broth at 37 °C.

Growth curve data were processed using the free DMFit curve-fitting software (v2.1; courtesy of the Institute of Food Research, Norwich, UK) and the function developed by Baranyi et al. [30]. EO mix MIC/2 was the EO mix concentration where we observed bacterial growth recovery. Mathematical analysis enabled the determination of three growth parameters: ODmax (maximal optical density) data, which gives the maximum increases in OD during incubation; maximum growth rate (μmax), which occurs in the early exponential growth phase; and lag time (Lag T), which is the time lapse before an OD increase as per Baranyi et al.’s function [30]. Extension of the lag time was calculated using the formula given in Hayouni et al. [31].

### 2.5. Statistical Processing

For each bacteria strain, the comparison of the three growth parameters between EO mix MIC/2 assays versus negative controls were analyzed in XLStat^®^ premium software v2023.3.1.1416 (Addinsoft, France). After performing the *t*-test (*p*-value > 0.05), we choose non-parametric Mann–Whitney and Kruskal–Wallis tests (0.05 significance level).

## 3. Results

### 3.1. Composition of the EO Mix

The EO mix was composed of 15 natural components. The main components were thymol (37.72%), carvacrol (26.76%), geranial (8.33%), β-citral (5.83%), and p-cymene (4.39%) (Table 1).

### 3.2. Comparison of the Microdilution and the Macrodilution Methods

EO mix MIC values were determined according to macrodilution and microdilution method for three out of four bacteria strains (Table 2). No EO mix MIC was determined for *P. aeruginosa.*

### 3.3. Growth Parameters of the Microdilution Method

Bacterial strains’ growth parameters, obtained with the microdilution method, are presented in Table 3.

#### 3.3.1. Growth Rate (µmax)

*B. cereus*, *E. coli*, and *S. aureus* exposed to EO mix exhibited a significant decrease in growth rate (*p* < 0.05) compared to their respective negative control (0.39 h-1 vs. 3.46 h-1 for *B. cereus*, 0.66 H-1 vs. 4.66 H-1 for *E. coli*, and 0.58 H-1 vs. 2.15 H-1 for *S. aureus*). For *P. aeruginosa*, no MIC could be determined, so no µmax could be calculated for sub-MIC. *P. aeruginosa* exhibited growth in all of the conditions tested. We were able to determine a µmax for *P. aeruginosa* in all of the conditions tested (see Appendix A). The µmax values of *P. aeruginosa* in technical replicates for each EO mix dilution were statistically compared against the controls (*P. aeruginosa* without EO mix): µmax values from dilution 7.81 × 10^2^ μg/mL to 1 × 10^5^ μg/mL were significantly different to µmax values of controls (*p* < 0.005).

According to the means of the biological replicates (Table 3), there was no significant difference between the µmax values of the three bacterial strains tested. However, *E. coli* tended to have higher µmax values than S. *aureus* and *B. cereus*.

#### 3.3.2. Lag Phase (LagT)

*B. cereus*, *E. coli*, and *S. aureus* exposed to EO mix exhibited a significantly higher lag time (*p* < 0.05) compared to their respective negative controls (0.63 h vs. 0.09 h for *B. cereus*, 0.55 H vs. 0.07 H for *E. coli*, and 0.63 H vs. 0.15 H for *S. aureus*) (Table 3). For *P. aeruginosa*, no MIC could be determined, so no lagT could be calculated at sub-MIC. There was no significant difference in *P. aeruginosa* lag phase between EO mix assays and controls without EO mix (see Appendix A).

According to the means of the biological replicates (Table 3), the lag phase was longer for *S. aureus* than *B. cereus* and *E. coli*, but the between-strain differences were not statistically significant.

#### 3.3.3. ODmax

After 24 h of the experiment, OD max values for *B. cereus*, *E. coli*, and *S. aureus* exposed to EO mix were significantly lower (*p* < 0.05) than their respective negative control OD max values (Table 3). OD max values for *B. cereus* and *E. coli* decreased more than two-fold (0.30 vs. 0.73 for *B. cereus* and 0.18 vs. 0.86 for *E. coli*; Table 3). For *P. aeruginosa*, no MIC was determined, so no OD max could be calculated at sub-MIC. *P. aeruginosa* exhibited growth in all of the conditions tested. *P. aeruginosa* OD max values for EO mix concentrations 1 × 10^5^ μg/mL, 1.25 × 10^4^ μg/mL, 3.13 × 10^3^ μg/mL, 1.56 × 10^3^ μg/mL, and 7.81 × 10^2^ μg/mL were significantly different to control OD max (*p* < 0.005; see Appendix A)

According to the means of biological replicates, OD max was higher for *B. cereus* than *S. aureus* and *E. coli*. OD max values were significantly different between *E. coli* and *B. cereus* (*p* < 0.005) but not between *E. coli* and *S. aureus* or between *B. cereus* and *S. aureus*.

### 3.4. Technical and Statistical Comparison of the Two MIC Methods

#### 3.4.1. Statistical Descriptive Parameters of the Results of the Microdilution Method

##### Comparison of the Technical Replicates for Lag T, µmax, and ODmax Parameters

Repeatability was described based on the standard deviation (SD) of technical replicates calculated for each growth parameter.

Negative controls (without EO mix):

For the negative controls of all growth parameters, the SD values for technical replicates ranged from 0.00 to 4.27 (Table 3).

For the lag phase of negative controls, SD values for technical replicates ranged from 0.00 to 0.05.

For the growth rate of negative controls, SD values for technical replicates ranged from 0.00 to 4.27.

For the OD max of negative controls, SD values for technical replicates ranged from 0.00 to 0.13.

The DMFit model showed a low SD for technical replicates of all negative controls except the *P. aeruginosa* strain.

Assays with EO mix:

For the lag phase of strains with EO mix, SD values for technical replicates ranged from 0.00 to 0.52.

For the growth rate of strains with EO mix, SD values for technical replicates ranged from 0.01 to 3.31. For the OD max of strains with EO mix, SD values for technical replicates ranged from 0.00 to 0.34.

The R^2^ values for growth-model technical replicates ranged from 31.48% to 99.50%. R^2^ values for growth-model technical replicates of positive controls ranged from 81.72% to 99.53%. R^2^ values for growth-model technical replicates of strains with EO mix ranged from 31.38% to 99.22%.

##### Comparison of the Biological Replicates for Lag T, µmax, and ODmax Parameters

Reproducibility was evaluated based on the SD of biological replicates calculated for each growth parameter.

Negative controls (without EO mix):

For the negative controls of all growth parameters, the SD values for biological replicates ranged 0.01 to 2.51 (Table 3).

For the lag phase of negative controls, SD values for biological replicates ranged from 0.01 to 0.03.

For the growth rate of negative controls, SD values for biological replicates ranged from 0.32 to 2.51.

For the OD max of negative controls, SD values for biological replicates ranged from 0.02 to 0.07.

Assays with EO mix:

For the lag phase of strains with EO mix, SD values for biological replicates ranged from 0.08 to 0.41.

For the growth rate of strains with EO mix, SD values for biological replicates ranged from 0.56 to 1.12.

For the OD max of strains with EO mix, SD values for biological replicates ranged from 0.09 to 0.26.

The R^2^ values for growth-model biological replicates ranged from 57.44% to 99.20%. R^2^ values for growth-model biological replicates of positive controls ranged from 92.50% to 99.20%. R^2^ values for growth-model biological replicates of strains with EO mix ranged from 57.44% to 87.90%.

#### 3.4.2. MIC Results of the Macrodilution Method

The reading of MIC results was determined by visible turbidity (Figure 1) and by a spectrophotometer set to 600 nm. The high turbidity due to the EO mix affected the OD measurement. The MHB + EO mix of each dilution was used as blank to determine MIC, but it was higher than 1 unit for concentrations between 1.25 × 10^4^ μg/mL and 1 × 10^5^ μg/mL, so the turbidity due to bacterial growth could not be not determined well by spectrophotometry readings at a single wavelength for the dilutions between 1.25 × 10^4^ μg/mL and 1.00 × 10^5^ μg/mL. However, for the following dilutions tested (i.e., from 9.77 × 10^1^ μg/mL to 6.25 × 10^3^ μg/mL), the OD of MHB + EO mix with MHB as blank was lower than 1 unit, making it possible to determine the turbidity due to bacterial growth based on spectrophotometry readings.

#### 3.4.3. Comparison of the MIC Results of the Macrodilution Method vs. the Microdilution Method

MIC values between the macrodilution method and the microdilution method were only significantly different for *B. cereus* (Table 1).

### 3.5. Evaluation of the Effect of DMSO on Bacterial Growth

We investigated several effects of DMSO concentration on bacterial growth using 24 h OD measurements (Figure 2).

DMSO at 5% concentration impacted the growth of all bacterial strains (see Figure 2A–D).

The growth rates of *P aeruginosa*, *E. coli*, and *S. aureus* were less impacted by DMSO at 3% (Figure 2A,C,D). The growth of *B. cereus* was not impacted compared to *B. cereus* control at 3% DMSO concentration (Figure 2B).

With DMSO 1%, the growth curves of *E. coli* and *S. aureus* were not different from their respective controls (Figure 2C,D). For *P. aeruginosa* with DMSO 1%, the growth curve exhibited the same OD max values as the control curve (up to 1.5 OD unit) (Figure 2A), but the exponential-phase growth rate was greater in the control condition (Figure 2A). DMSO impacted all bacterial growth rates in a concentration-dependent manner.

## 4. Discussion

### 4.1. Statistical and Technical Points

The EO mix MIC values found in this study had SDs from 65.3 to 795 for MIC values ranging from 217 µg/mL to 1390 µg/mL. Visual MIC determination can be difficult with the macrodilution method, especially with a colored antibacterial solution such as the EO mix. This method would likely lead to different results between operators and biological replicates. Coloration and incomplete dissolution of the EO mix could also lead to increasing SDs with OD reads. Both dilution methods showed limits for MIC determination with colored antibacterial substances. Nevertheless, the automated microdilution method allowed the calculation of additional bacterial growth parameters such as growth rate, lag phase, and maximal OD value. DMFit was suitably able to determine the MIC and to give further information on strain growth in the presence of EO mix. The high R^2^ from positive controls confirmed the robustness of the DMFit model in these cases. Maximal OD value and lag phase parameters were efficient when there was significant growth (µmax > 0). MIC values were only significantly different between the macrodilution method and the microdilution method for *B. cereus*. Several technical and/or biological factors could explain this difference. *B. cereus* is a spore-forming, anaerobic, facultative, rod-shaped bacterium. The type of metabolism may impact growth, and thus the MIC of *B. cereus* depends on whether it is grown on the surface of an agar medium or in a tube in a broth medium. In previous experiments, we showed that *B. cereus* strains did not use the same metabolism for growth and are able to produce high concentrations of ethanol by metabolizing excess carbon [32]. *B. cereus* could potentially react differently to the EO mixture when it is grown in a microplate well than when it is grown in the microtube, as the microtube has a greater anaerobic zone. Further studies could include additional morphological bacterial cells checks. We suggest using flow cytometry or epifluorescent microscopy with FITC filter to assess the proportions of the vegetative cells and spores. The turbidity due to bacterial growth was not as visible as the color due to the EO mix, especially for dilutions between 1/10 and 1/80. The same problem occurred with the spectrophotometer at a single wavelength; i.e., the color of the EO mix disturbed the OD measurement in dilutions from 1/10 to 1/80 [33,34]. The Bioscreen C reader measured optical density (OD) in the 420–580 nm wavelength band and was able to detect bacterial growth more easily despite the high color of the EO mix. It thus enabled us to monitor *P. aeruginosa* growth in all dilutions tested and to determine all of the corresponding growth parameters. DMSO was used in this study to solubilize the EO mix in MHB. DMSO impacted the growth of all model bacteria studied in a concentration-dependent manner (from 1% to 5%). The bacterial MICs found were from 217 to 1390 µg/mL for *E. coli*, *B. cereus*, and *S. aureus.* In these experimental conditions, DMSO proportions were between 0.001% and 0.008%; the effect of DMSO on bacterial growth did not affect the MIC results. We suggest testing the impact of other emulsifiers used to solubilize EOs (such as Tween 20, Tween 80) before experiments. The literature has identified other technical parameters that can affect MIC results [3,35,36]. The adherence of EOs to the polypropylene pipette tips could also affect the results [37]. The emulsion of EOs was critical, and agar microdilution, sonication, and/or the use of emulsification agents such as DMSO, Tween 80, or Tween 20 were proposed to improve it. Hood et al. (2003) [38] established an optimized broth dilution method, using 0.02% Tween 80 to emulsify the oils, and showed that it is as the most accurate method for testing the antimicrobial activity of hydrophobic and viscous EOs. The insolubility of EOs causes variability and instability of the OD measurement and the mixture [39]. With the microdilution method, we observed a decrease in OD during the first 2 h that probably reflected partial sedimentation of the EO mix, but this phenomenon has no impact on MIC results. Zgoda et al. (2001) [40] first observed this phenomenon when they mixed natural plant extracts with sterile water to make up a 2.5% DMSO/water/extract solution and found that the extracts precipitated from the solution. To overcome this problem, 100 mg of solid extract were prepared from aliquots on a separate polypropylene 96-well plate. The extracts were then dissolved in pure DMSO at an appropriate concentration and added to 95 µL sterile water on a sterile 96-well bioassay plate. We compared the MIC results after runs with continuous shaking vs. discontinuous shaking and found no difference between the MIC results. This was not in agreement with Vanegas et al. (2021) [27], who observed that constant agitation guaranteed continuous contact of oil with bacteria and improved MIC results. This divergence between findings may be due to one of two reasons. First, we used DMSO, whereas Vanegas et al. (2021) [27] used Tween 80, which could be more efficient in terms of producing an emulsion. Second, the constant agitation may not have been high enough to limit sedimentation and thus did not modify our MIC results. Donadu et al. (2021) [41] and Li et al. (2015) [26] used Tween 80 in broth and Tween 20 in agar with concentrations of 0.5% (*v*/*v*) and observed good dispersion of the oils in liquid medium. Tan et al. (2015) [34] reported that the addition of Tween 80 may be a limitation to using solvents, and that the concentration of Tween 80 used could tend to be low, ranging from “a few drops” to 0.02% and up to 5% for less-polar polyphenols like flavonoids. Oils may require up to 10% Tween 80. Ghosh et al. (2013) [42] applied sonication to the emulsion for 30 min and observed a reduction in droplet size and an incremental increase in the stability of the emulsion. However, Chen et al. (2023) [39] found that the agar dilution method had better repeatability than microbroth dilution, but some operations caused some deviation of repeatability for agar dilution. They concluded that the microbroth dilution method was suitable for determining MICs for EOs. Tan et al. (2015) [34] added that the agar dilution assay could be an alternative to broth microdilution for testing less-polar natural extracts. Chen et al. (2023) [39] showed that different EOs have different solvent requirements for tests against different bacteria, so the limit of the solvents should be evaluated before use. Here, we tested graduated concentrations of DMSO on the target strains and found that, in our experimental conditions, between 0.001% and 0.008% DMSO had no impact on the growth of the bacterial strains, and that DMSO could be used to dissolve the EO mix. EOs are hydrophobic and have high viscosity. These properties may reduce dilutability or cause unequal distribution of the oil through the medium, which would lead to separation of the oil–water phases [43] and the formation of micelles [44]. The antimicrobial activity of EOs could be related to their ability to form micelles, which is a microbially inactive form [45]. In contrast, Man et al. (2019) [46] showed that some formulas of colloid or micelle suspensions of EOs such as oregano, thyme, or lemon oil can have some antimicrobial activity. This suggests that size of the micelles is a critical factor for the antimicrobial action of EOs. Man et al. (2019) [46] proposed a protocol for preparing EOs in solution without emulsion solvents and only with water mixed overnight and sonicated. The volatility of the EOs tested is a factor that could affect the MIC results. Indeed, EOs are very complex mixtures of volatile components, and such long incubation times may result in evaporation or decomposition of some of the components during the testing period, which makes it important to set up specific experimental conditions to avoid these phenomena [47]. It may also be useful quantify the active EOs’ residues at the end of the incubation period, which could serve to evaluate the proportion of EOs potentially complexed with the culture medium. The culture medium is also a factor that could significantly affect the MIC results. In this study, we used MH broth because it is the medium most commonly used for broth and agar dilutions for the standardized (internationally accepted) method for the MIC determination of natural substances. However, Hulankova (2022) [48] compared BHI, TSB, and MHB and concluded that MHB is far from ideal for determining the antimicrobial properties of EOs. EOs’ MIC values were significantly lower in MHB for all pathogens. This was probably due to an interaction of EOs’ components with starch and to the reduced ability of the bacteria to repair cell damage in a nutrient-deficient medium. TSB seems to be the most suitable candidate reference medium for any future standard broth dilution method for EOs antimicrobial testing. The literature has clearly identified the most important factors causing variance in MIC between studies, including incubation conditions, culture media, and the use of emulsifiers or solvents. Balouiri et al. (2016) [49] also mentioned other factors, such as inoculum size and the end-points determined. Several authors have shown that bacterial inoculum size affects MIC determination. Koutsoumanis et al. (2005) [50] and Lambert et al. (2000) [45] showed that the level of inhibitor factor decreases with the size of the inoculum. This suggests that MICs may depend on the concentration of the bacterial target encountered in real situations [51]. Based on our results, the weak R^2^ and high SD between the technical replicates and biological replicates for each bacterial strains’ MIC/2 could be explained by variability due to all of the factors listed above, but inoculum size could also play a major role. The heterogeneity of bacterial physiological state in the inoculum could also have an impact on the growth/no-growth boundary [52]. Tan et al., 2015 [34] added that regardless of the inoculum size used, the bacterial suspension should be used within 30 min after it has been adjusted, to avoid significant changes in cell numbers. Of course, when the growth/no-growth boundary is examined for the determined MIC, it is also possible to calculate the non-inhibitory concentration (NIC). Lambert et al. (2000) [45] defined NIC as the smallest concentration of inhibitory substance that observably slows normal growth. NIC is only observable with the automated method, and it is an important indicator for evaluating the stage and timing of the bacteriostatic effect. In the supplementary data, we showed that the EO mix had an impact on the growth rate (µmax) of *P. aeruginosa* but not on the duration of its lag phase. In addition, the growth of ps. aeruginosa in the presence of EOs at dilutions from 1/10 to 1/320 was similar to that of the control without EOs. Its growth rate decreased from 1/320th dilution of the EOs mixture. This study made it possible to determine the MIC of EOs via a reproducible and convenient high-throughput, time- and cost-effective method, as well as limit the measurement uncertainty of MIC by microdilution method compared to macrodilution method. Moreover, we provided tools for characterizing the action of EOs on bacterial growth, especially at sub-MIC EOs concentrations. The combination of lag-phase duration, µmax, and OD max values effectively detected growth or no-growth and identified which growth parameter(s) were affected by the EOs. Future experiments could focus on validation of the microdilution method on an anaerobic bacterial model. We also intend to explore other screening methods to prevent color issues of the EO mix and plant extracts.

### 4.2. Antibacterial Activity of the EO Mix

The EO mix showed activity against three of the four model bacteria strains tested. Its composition was a blend of thymol, carvacrol, citral, and p-cymene. Thymol and carvacrol, two phenolic isomers, have strong antibacterial activity against several bacterial strains [3,53,54]. At sub-MIC, carvacrol impacted both *E. coli* and *B. cereus* growth-curve kinetics, which translated into lower OD max values [55]. These results are consistent with our study. In contrast, for calculated growth parameters, such as bacteriostatic effect, Pedreira et al. (2024) [55] found a higher susceptibility to carvacrol for *E. coli* than *B*. *cereus*. Thymol and carvacrol have mainly been studied for their action against *P. aeruginosa* by inhibiting biofilm adherence and formation [56,57]. Maggini et al. (2017) [58] found MIC values over 0.5% (*v*:*v*) for *P. aeruginosa* exposed to oregano oil, with an oregano oil composition of carvacrol >70% and thymol 21.5%. Given the composition of the EO mix studied here (see Section 3.1), the oregano MIC found by Maggini et al. (2017) [58] was much higher than for the strongest EO mix concentration that we could have tested in our study, which explains why we were unable to find an EO mix MIC against *P. aeruginosa*. Citral also has well-known antibacterial effects [59,60,61]. Citral inhibits bacterial growth in a species-dependent manner: *B. cereus* was shown to be more susceptible to lemongrass containing more than 70% citral compared to *E. coli* or *S. aureus* [62,63]. Furthermore, gram-positive bacteria were found to be more susceptible to lemongrass than gram-negative bacteria [64]. These results are consistent with our study, as the gram-positive bacteria (*B. cereus* and *S. aureus*) showed higher susceptibility to the EO mix than the gram-negative *E. coli* (Table 1). Our study is also consistent with the conclusions of Murbach et al. (2014) [65]. Nevertheless, we were unable to determine MIC values for *P. aeruginosa*. *P. aeruginosa* is known to demonstrate greater tolerance to volatile compounds than other gram-negative bacteria such as *E*. *coli* [66]. Note too that among several volatile compounds tested by Cox and Marckham (2007) [66], *P. aeruginosa* was susceptible to trans-cinnamaldehyde, which was absent from the EO mix tested here. The absence of exposure to trans-cinnamaldehyde could explain why *P. aeruginosa* showed no growth and gave no MIC here. Moreover, some studies have demonstrated *P. aeruginosa*’s ability to counteract cinnamaldehyde’s antibacterial activity and other terpenes’ action using a pump efflux mechanism [67,68]. This remarkable *P. aeruginosa* ability to fight against antibacterial volatile compounds, even in the presence of the most powerful antibacterial molecules such as cinnamaldehyde, showed that we could not expect to find a *P. aeruginosa* MIC in the absence of cinnamaldehyde in the EO mix. In addition, Naik et al. (2010) [64] found no susceptibility of *P. aeruginosa* to citral-rich lemongrass. Interestingly, Liu et al. (2021) [69] showed that thymol inhibits *P. aeruginosa* growth by permeabilization of the outer and inner membrane at an MIC of 0.25 mg/mL. The fact that we failed to find any MIC for *P. aeruginosa* in this experiment may be due to the fact that the EO mix only contained 37.29% thymol. However, exposure to the EO mix from 0.005 μg/mL to 5 μg/mL affected *P. aeruginosa* growth parameters in a heterogeneous way. Growth parameters showed lag phases of *P. aeruginosa* from 0.15 ± 0.04 H to 0.42 ± 0.09 H (Appendix A) and OD max values that varied from 0.99 ± 0.11 to 1.25 ± 0.10. However, the lag-phase values and OD max values were not always significantly different (*p* < 0.05) from control values, and differences were not proportional to EO mix dilutions. This suggested a heterogeneous adaptation of *P. aeruginosa* to different EOs. Nevertheless, all concentrations of the EO mix led to a decrease in growth rate (from 4.01 ± 0.60 H-1 to 1.03 ± 0.20 H-1) compared to the control growth rate (7.51 ± 2.51 H-1). This proved that the EO mix was able to reduce *P. aeruginosa* growth at all concentrations even if its antibacterial effect was not enough to completely inhibit bacterial growth. Consequently, we were unable to determine an MIC for *P. aeruginosa*. *P. aeruginosa* is well known to be able to adapt and survive to various environmental stressors via a broad range of resistance mechanisms, often present in the same clinical isolate [70]. *P. aeruginosa* can mobilize molecular mechanisms, efflux pumps, transcription regulation, and quorum sensing in various ways to counteract the effects of antibacterial drugs. These capabilities confer *P. aeruginosa* with various potential strategies for surviving against EOs, even when they are blended with various other molecules. The modes of action of various EOs against various bacterial strains are well documented, but little is known about the complex effects of EO mixes on bacterial cells. In previous work using flow cytometry, we demonstrated the mode of action of another EO blend (similar to the EO mix here but containing cinnamaldehyde) against gram-negative *E. coli* [16]. The association of about 9% thymol and 8% carvacrol in the mix demonstrated bacteriostatic activity against *E. coli* by membrane permeabilization and pH perturbations. Thymol and carvacrol are known to inactivate bacterial enzymes and proteins, and they also have a global effect on DNA and protein synthesis at sub-MIC levels [71]. This mode of action could also explain the decrease in bacterial growth rate found with sub-MIC EO mix in this study (Table 3), as DNA and protein synthesis are key for cell growth. Furthermore, the sustained loss of ions or metabolites due to exposure to an EO can compromise microbial metabolism and lead to cell death [72]. This same mechanism may have led to the reduced growth rate and OD max values found here. Thymol and citral have been reported to target bacterial membranes and disrupt bacterial homeostasis in a gram-negative strain [73,74]. In particular, exposure to thymol led to cell membrane depolarization, decreased intracellular ATP concentrations, and lower pH. Cell membrane depolarization was also studied in our previous work exposing *E. coli* to a complex EO mix [16] and confirmed thymol action on cell membrane depolarization. Several terpenes such as thymol, carvacrol, and p-cymene targeted membrane integrity in another way against a gram-negative strain: the effect reported is an increase in the level of reactive oxygen species (ROSs) [75]. ROSs are known to cause lipid peroxidation in cell membranes, which affects membrane permeabilization. This mechanism could also explain the decrease in OD max with exposure to the EO mix compared to control (Table 3), as OD max is directly correlated with bacterial concentration.

This study focused on the antibacterial activity of the EO mix rather than individual EO components. This choice was motivated by the fact that it would be too hard to unravel the complex synergistic, antagonist, and/or additive effects of interactions between individual EO components [76], even with different essential oils [77]. There is increasing evidence of synergism between major and minor components within one EO but also between the major components themselves [78]. EO blends are expected to have synergies that will exert stronger antibacterial effects than individual EOs or their individual components [71,79]. In particular, thymol, carvacrol, and eugenol at different combinations exhibit synergism or moderate synergism depending on their respective ratios [80,81]. In general, phenolic monoterpenes increase the bioactivities of other components in mixtures [82]. In particular, thymol and carvacrol exhibit antibacterial synergism or additive antibacterial effects against gram-negative strains including *E. coli*, *S. aureus*, and *P. aeruginosa* [45,83]. A blend of carvacrol and p-cymene, which was one of the components of the EO mix studied here, exhibited synergistic effects against *B. cereus* [84], as did oregano oil associated with thymus oil [85]. Our study is consistent with this literature, as we found MIC values for these same bacterial strains with our EO mix composed of these related individual oils. This study points to the utility of exploring the activity of complex EO blends against several pathogenic bacterial strains.

## 5. Conclusions

This study represents one of the first attempts to specify the statistical parameters of a microdilution method to determine bacterial growth parameters (OD max, lag phase, and growth rate) and the MIC of an EO mix. Our experiments clarified key technical specifications for determining the MIC of an EO mix. Our study emphasizes the importance of ensuring the accuracy of MIC results by performing three technical and three biological replicates using an automated system for accuracy and combining three growth parameters (OD max, lag phase, and growth rate) to assess the impact of the EO mix, particularly at sub-MIC levels. We validated the in vitro antibacterial action on *E. coli*, *B. cereus*, *S. aureus*, and *P. aeruginosa* of a specific EO mix that did not contain cinnamon oil, which is an oil known to have strong antimicrobial action. This work thus contributes to efforts to study the antibacterial effects of EOs via a robust and repeatable in vitro method, combined with the growth parameters for screening pathogenic bacteria. This effort could be crucial to help address the major global health challenges caused by antimicrobial resistance.

## Figures and Tables

**Figure 1 microorganisms-13-01125-f001:**
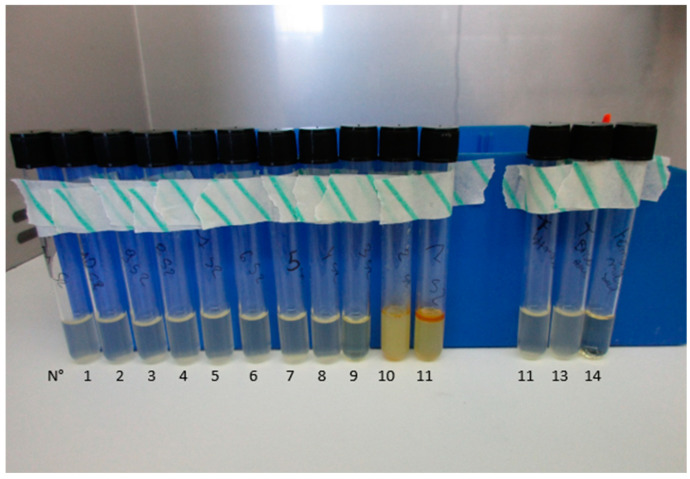
Macrodilution test tubes of EO mix at progressive 2:1 ratio dilutions against *E. coli*; EO mix concentrations are as follows in numerated tubes: n°1 9.77 × 10^1^ μg/mL, n°2 1.95 × 10^2^ μg/mL, n°3 3.91 × 10^2^ μg/mL, n°4 7.81 × 10^2^ μg/mL, n°5 1.60 × 10^3^ μg/mL, n°6 3.10 × 10^3^ μg/mL, n°7 6.20 × 10^3^ μg/mL, n°8 1.25 × 10^4^ μg/mL, n°9 2.50 × 10^4^ μg/mL, n°10 5.00 × 10^4^ μg/mL, n°11 1.00 × 10^5^ μg/mL; tube n°11 is negative control (MHB+ *E. coli*), tube n°12 is *E.coli* + DMSO 1% + MHB, tube n°14 is MHB.

**Figure 2 microorganisms-13-01125-f002:**
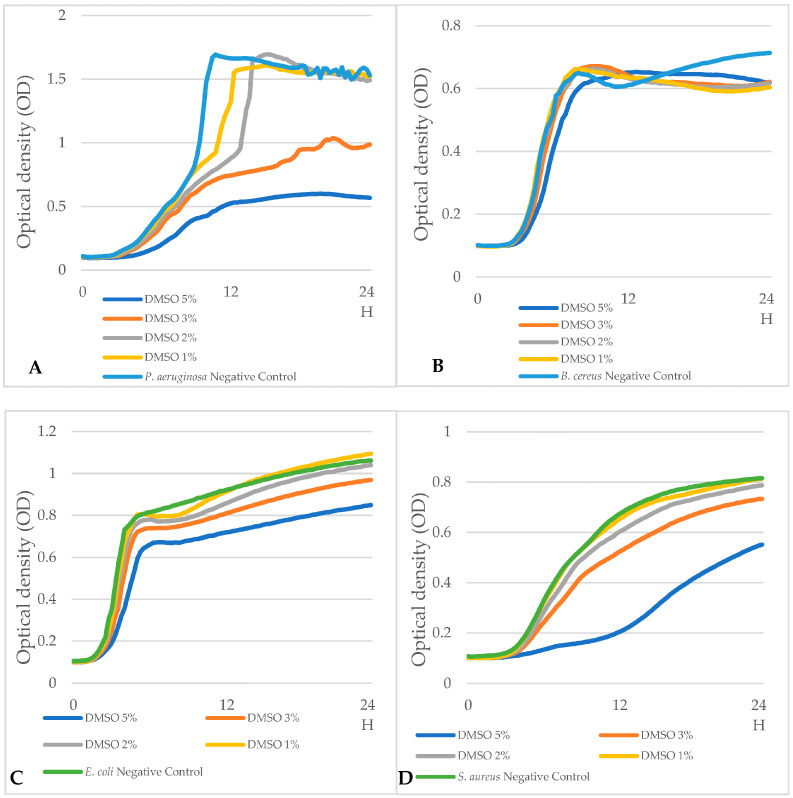
Illustrations of 24 h growth curves (OD) of (**A**) *P. aeruginosa*, (**B**) *B. cereus*, (**C**) *E. coli*, and (**D**) *S. aureus* at different DMSO concentrations (5%, 3%, 2%, 1%).

**Table 1 microorganisms-13-01125-t001:** EO mix composition analyzed by GC-MS.

Compound Name	Rate (%)
α-pinene	0.54
Camphene	0.23
p-cymene	4.39
Limonene	0.40
γ-terpinene	2.15
Diallyl disulfide	0.42
Linalol	0.30
Isopulegol	2.10
Citronellal	0.70
Neoisopulegol	0.31
Terpinene-4-ol	0.43
α-terpineol	0.20
Citronellol	0.80
β-citral	5.83
Geraniol	2.63
α-citral	8.01
Thymol	37.72
Carvacrol	26.76
Dially trisulfide	0.69
Caryophyllene	0.97

**Table 2 microorganisms-13-01125-t002:** Comparison of the EO mix MIC (µg/mL) determined by the microdilution and macrodilution methods for *E. coli*, *B. cereus*, *P. aeruginosa*, and *S. aureus* strains.

Bacterial Strain	Macrodilution Method EO Mix MIC (µg/mL)	Microdilution Method EO Mix MIC (µg/mL)
*Bacillus cereus*	6.08 × 10^2^ ± 2.06 × 10^2^ a	2.17 × 10^2^ ± 6.53 × 10^1^ b
*Pseudomonas aeruginosa*	N.D.	N.D.
*Staphylococcus aureus*	3.92 × 10^2^ ± 2.41 × 10^2^ a	3.69 × 10^2^ ± 6.53 × 10^1^ a
*Escherichia coli*	1.39 × 10^3^ ± 7.59 × 10^2^ a	5.21 × 10^2^ ± 1.95 × 10^2^ a

N.D: not determined. Values are means of nine replicates (three biological replicates and three technical replicates in one biological replicate); values in a row with the same letter are not significantly different (*p* > 0.05). MIC values were not significantly different between the macrodilution and microdilution methods (*p* > 0.05) except for the *B. cereus* strain (Table 1). MIC values were between 2.17 × 10^2^ µg/mL and 1.39 × 10^3^ µg/mL (Table 1). Bacterial strains showed different susceptibilities to the EO mix: *S. aureus* was the most EO-mix-susceptible strain (EO mix MIC between 3.69 × 10^2^ µg/mL and 3.92 × 10^2^ µg/mL), and *E. coli* had the highest EO mix MIC (EO mix MIC between 5.21 × 10^2^ µg/mL and 1.39 × 10^3^ µg/mL).

**Table 3 microorganisms-13-01125-t003:** Growth parameters (µmax, lagT, and OD max) of *E. coli*, *B. cereus*, *P. aeruginosa*, and *S. aureus* cultured with EO mix at progressive 2:1 dilution. Values with different letters between two growth conditions of the same bacterial strain are significantly different (*p* < 0.05). ‘Negative control’: bacterial strain in MHB (Mueller–Hinton broth); * mt: mean of three technical replicates; ** mb: three biological replicates; N.D.: not determined.

Bacterial Strains Growth Conditions (With or Without EO Mix)	EO Mix MIC/2 (µg/mL)	Lag Phase (LagT) (h)	Growth Rate (µmax) (h-1)	OD Max (uOD)	R^2^	Bacterial Strains Growth Conditions (With or Without EO Mix)	EO Mix MIC/2 (µg/mL)	Lag Phase (LagT) (h)	Growth Rate (µmax) (h-1)	OD Max (uOD)
	mt *	mb **	mt	mb	mt	mb	mt	mb	mt	mb
*Bacillus cereus* negative control	mt1	-	0.09 ± 0.01	0.09a ± 0.03	3.73 ± 0.34	3.46a ± 0.90	0.74 ± 0.00	0.73 ± 0.04	81.72%	92.50%
	mt2		0.11 ± 0.00		4.27 ± 0.02		0.68 ± 0.01		99.50%	
	mt3		0.06 ± 0.02		2.37 ± 0.31		0.77 ± 0.01		96.31%	
*Bacillus cereus* with EO mix	mt1	108.39 ± 32.67	0.67 ± 0.52	0.63b ± 0.41	0.05 ± 0.04	0.39b ± 0.56	0.11 ± 0.00	0.30b ± 0.14	91.59%	87.90%
	mt2		0.25 ± 0.07		1.11 ± 0.29		0.39 ± 0.03		77.75%	
	mt3		0.97 ± 0.00		0.02 ± 0.01		0.11 ± 0.00		93.49%	
	mt1	-	0.08 ± 0.00	0.07a ± 0.01	5.36 ± 0.26	4.66a ± 0.63	0.86 ± 0.01	0.86a ± 0.02	98.04%	97.50%
*Escherichia coli* negative control	mt2		0.06 ± 0.00		4.42 ± 0.41		0.87 ± 0.02		96.46%	
	mt3		0.06 ± 0.01		4.19 ± 0.47		0.86 ± 0.04		97.00%	
*Escherichia coli* with EO mix	mt1	260.50 ± 97.50	0.27 ± 0.25	0.55b ± 0.08	2.02 ± 3.31	0.66b ± 1.12	0.33 ± 0.27	0.18b ± 0.09	89.90%	82.67%
	mt2		0.97 ± 0.00		(-)0.01 ± 0.04		0.10 ± 0.00		70.18%	
	mt3		0.40 ± 0.00		(-)0.03 ± 0.03		0.10 ± 0.00		87.93%	
*Pseudomonas aeruginosa* negative control	mt1	-	0.23 ± 0.02	0.03 ± 0.02	7.22 ± 0.56	7.51 ± 2.51	1.55 ± 0.01	1.62 ± 0.07	98.00%	98.00%
	mt2		0.24 ± 0.04		9.41 ± 4.27		1.63 ± 0.07		98.00%	
	mt3		0.21 ± 0.01		5.92 ± 0.75		1.68 ± 0.05		98.00%	
*Pseudomonas aeruginosa* with EO mix	mt1	ind	ind	ind	ind	ind	ind	ind	ind	ind
	mt2		ind		ind		ind		ind	
	mt3		ind		ind		ind		ind	
*Staphylococcus aureus* negative control	mt1	-	0.17 ± 0.05	0.15a ± 0.03	2.37 ± 0.61	2.15a ± 0.32	0.89 ± 0.13	0.94a ± 0.07	99.53%	99.20%
	mt2		0.14 ± 0.01		2.04 ± 0.00		0.96 ± 0.01		99.02%	
	mt3		0.14 ± 0.01		2.04 ± 0.05		0.97 ± 0.02		99.06%	
*Staphylococcus aureus* with EO mix	mt1	184.61 ± 32.67	0.18 ± 0.11	0.63b ± 0.41	1.51 ± 1.29	0.58b ± 0.975	0.50 ± 0.34	0.25b ± 0.26	99.22%	57.44%
	mt2		0.74 ± 0.39		0.23 ± 0.40		0.16 ± 0.09		41.63%	
	mt3		0.97 ± 0.00		(-)0.00 ± 0.01		0.11 ± 0.00		31.48%	

ind: indeterminate. The different letters indicate significant difference (*p*-value < 0.05) per strain between negative control (without EO mix) and with EO mix.

## Data Availability

The original contributions presented in this study are included in the article/Appendix A. Further inquiries can be directed to the corresponding author.

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
