# Peer review of "Comparison of In Vitro Methods for Assaying the Antibacterial Activity of a Mix of Natural Essential Oils Against Zoonotic Bacteria"

_microorganisms, 2025, doi:10.3390/microorganisms13051125_

Round 1

Reviewer 1 Report

Comments and Suggestions for Authors

Manuscript Title

“Comparison of in vitro methods for assaying the antibacterial activity of a mix of natural essential oils against zoonotic bacteria”

Summary of the Manuscript

This manuscript addresses an important methodological question: how best to determine minimal inhibitory concentrations (MIC) of hydrophobic, volatile essential oils (EO) against common zoonotic bacteria. The authors compare broth macrodilution (tube-based) with an automated broth microdilution method (Bioscreen C), using four representative bacterial strains (Escherichia coli, Bacillus cereus, Pseudomonas aeruginosa, and Staphylococcus aureus). They additionally assess how sub-MIC concentrations of a blended EO formulation influence key bacterial growth parameters (lag phase, maximal growth rate, and maximal optical density).

Overall, the manuscript provides valuable data and highlights the practical concerns (e.g., color interference, partial miscibility, replicate variability) that researchers face when testing essential oil formulations. The topic is timely, given the rising interest in novel antimicrobial alternatives.

General and Major Comments

Relevance and Strengths

The study is valuable for laboratories seeking to optimize and standardize methods for determining MICs of natural products, especially essential oils.

Comparing two standard methods (macro- vs. microdilution) provides direct insights into best practices and potential pitfalls.

Essential Oil Composition and Identification

Key Revision Needed: The Methods section indicates that the essential oil components were identified with gas chromatography–mass spectrometry (GC–MS). However, the paper must include a compositional table listing all identified constituents with clear numerical values (in percentages or other appropriate units), explicitly stating the methods of identification (e.g., library match, retention index comparison, co-injections) and specifying the retention indices (RI) for each component.

If co-injections were performed to confirm the identity of major compounds, please note this.

If mass spectra were compared against libraries (e.g., NIST), the percentage of match or other relevant criteria should be briefly reported.

Statistical Rigor and Decimal Places

The manuscript presents a number of parameters (MIC values, growth rates, etc.) with means and standard deviations. However, there are places in the Results and Tables where the precision (number of decimal places) appears excessively high, which raises questions about the practical meaning and the rules of significant figures.

Key Revision Needed: Please ensure that reported values are rounded consistently and in accordance with statistical conventions. Using too many decimal places can imply an unwarranted level of precision. For each parameter (e.g., MIC values, growth rates, OD), it is typically sufficient to display two or at most three significant figures or decimal places, depending on the measurement’s variability.

Methodological Clarity

The discussion on color interference and phase separation in the macrodilution approach is thorough. However, it would be helpful if the authors explicitly stated any additional troubleshooting steps, such as how they corrected for or subtracted background absorbance in highly colored wells.

  1. cereus results: The MIC differences between macro- and microdilution for B. cereus are notable. A more explicit rationale for these differences—beyond sporulation or partial oxygen limitations—could help interpret these findings. For instance, mention whether vegetative cells or spores predominated, whether the authors performed additional morphological/spore checks, etc.

Sub-MIC Effects on Pseudomonas aeruginosa

The data indicate that P. aeruginosa grows in all EO concentrations tested, though some partial inhibition is observed in growth rate. Expanding on how P. aeruginosa’s known efflux mechanisms or its high intrinsic resistance might influence the inability to find an outright MIC is worthwhile. If cinnamaldehyde is typically needed for stronger activity and is absent here, mentioning that gap also helps contextualize why P. aeruginosa remains relatively unaffected.

Broader Recommendations

Since EO solubility is a known challenge, the authors might suggest future experiments comparing DMSO to other surfactants (e.g., Tween 20, Tween 80) or emulsification techniques.

A short paragraph clarifying how normality was (or was not) assessed before conducting Mann–Whitney/Kruskal–Wallis tests would help readers trust the statistical approach.

Minor Comments and Editorial Points

Figures and Tables

Ensure all axes and legends are clear and uniform in style.

Check that Table fonts and headings are consistent and that the italicization of bacterial species names is maintained throughout.

Abbreviations

Introduce “EO mix” once and use it consistently.

Spell out DMSO, MHB, etc., at first mention.

Reference Format

Verify that all references meet the journal’s stylistic requirements (Microorganisms typically uses MDPI style with complete author lists, DOIs, etc.).

Overall Recommendation

Recommendation: Minor Revision

This study contributes valuable methodological guidance for researchers investigating essential oils as antimicrobial agents. The manuscript is well-structured and the experiments are carefully performed, but the authors need to:

Include a comprehensive compositional table of the tested EO with explicit details (retention indices, means of identification, co-injections, etc.).

Adhere more strictly to statistical reporting conventions, particularly regarding the appropriate number of significant figures.

Provide brief clarifications on color interference methods, B. cereus spore/vegetative state checks, and a little more detail on the P. aeruginosa partial inhibition.

Once these points are addressed, the manuscript should be suitable for publication in Microorganisms.

Author Response

comment 1:

Key Revision Needed: The Methods section indicates that the essential oil components were identified with gas chromatography–mass spectrometry (GC–MS). However, the paper must include a compositional table listing all identified constituents with clear numerical values (in percentages or other appropriate units), explicitly stating the methods of identification (e.g., library match, retention index comparison, co-injections) and specifying the retention indices (RI) for each component.

If co-injections were performed to confirm the identity of major compounds, please note this.

If mass spectra were compared against libraries (e.g., NIST), the percentage of match or other relevant criteria should be briefly reported.

Response 1: the complete GC-MS analysis is currently ordered to ours internal laboratory, but we considered all your comments about this section, we won't be able to send you in time the entired data because of the laboratory planning.

comment 2: Key Revision Needed: Please ensure that reported values are rounded consistently and in accordance with statistical conventions. Using too many decimal places can imply an unwarranted level of precision. For each parameter (e.g., MIC values, growth rates, OD), it is typically sufficient to display two or at most three significant figures or decimal places, depending on the measurement’s variability.

response 2: thank you for your comment, all data were modified considering your recommandations

comment 3: The discussion on color interference and phase separation in the macrodilution approach is thorough. However, it would be helpful if the authors explicitly stated any additional troubleshooting steps, such as how they corrected for or subtracted background absorbance in highly colored wells.

response 3: we added the technical precision from Bioscreen C manual in the manuscript to answer to your question.

comment 4: cereus results: The MIC differences between macro- and microdilution for B. cereus are notable. A more explicit rationale for these differences—beyond sporulation or partial oxygen limitations—could help interpret these findings. For instance, mention whether vegetative cells or spores predominated, whether the authors performed additional morphological/spore checks, etc.

response 4: thank you for comment, we added the recommandation further studies: 

We suggest using flow cytometry or epifluorescent microscopy with FITC filter.(see line 416)

comment 5:

Sub-MIC Effects on Pseudomonas aeruginosa

The data indicate that P. aeruginosa grows in all EO concentrations tested, though some partial inhibition is observed in growth rate. Expanding on how P. aeruginosa’s known efflux mechanisms or its high intrinsic resistance might influence the inability to find an outright MIC is worthwhile. If cinnamaldehyde is typically needed for stronger activity and is absent here, mentioning that gap also helps contextualize why P. aeruginosa remains relatively unaffected.

response 5: thank you for your comment, we added the precision in the manuscript line 581

comment 6: Since EO solubility is a known challenge, the authors might suggest future experiments comparing DMSO to other surfactants (e.g., Tween 20, Tween 80) or emulsification techniques.

response 6: we added the suggestion for authors line 430

comment 7: A short paragraph clarifying how normality was (or was not) assessed before conducting Mann–Whitney/Kruskal–Wallis tests would help readers trust the statistical approach.

response 7: we added the precision line 180

comment 8: Figures and Tables

Ensure all axes and legends are clear and uniform in style.

Check that Table fonts and headings are consistent and that the italicization of bacterial species names is maintained throughout.

response 8: we modified the name of Y axis and the police of the legend in the figures and the italicization of bacterial species names.

comment 9: Abbreviations

Introduce “EO mix” once and use it consistently.

Spell out DMSO, MHB, etc., at first mention

response 9: al the precisions were added respectively at line 155 (DMSO), line 133 (MHB), line 162 (OD), line 171 (ODmax)

comment 10: 

Reference Format

Verify that all references meet the journal’s stylistic requirements (Microorganisms typically uses MDPI style with complete author lists, DOIs, etc.).

response 10: we downloaded the journal stylistic requirements from MDPI website to import in Zotero software; all the reference were generated by zotero according to Microoganisms reference style. we cited a book which don't have DOI and neither the CLSI norms.

Reviewer 2 Report

Comments and Suggestions for Authors

Dear authors,
its study is very interesting and has the potential to serve as a protocol for future research. however for that it must be very clear (to be replicable). 

The writing of the methodology needs to be improved (didactic so that it can be easily replicated). the results are not clear as presented here (it does not have structure, clarity of a scientific document). The discussion should be rewritten by better selecting literature to discuss its results. Here in this section there is a lot of repetition of the results and not the explanation of them "why"

most details see in the attached document,

Comments on the Quality of English Language

see my suggestion inside document.

Author Response

Thank you for your comments; we directly answered on your pdf file. Please, you could find in attach the new version of the manuscript.

Best  regards, 

Reviewer 3 Report

Comments and Suggestions for Authors

Dear authors,

I have reviewed your manuscript. In my opinion, the expectations of the reader based on the title of the study are not met by the content of the study. Thematically, the study addresses a useful topic, as natural substances are indeed a current focus of interest, and methodologies for their evaluation are not always entirely optimal. However, the text, in its current form, and the interpretation of the data are unsuitable for publication. The text needs to be better organized, with all inconsistencies and inappropriate formulations thoroughly corrected, particularly the MM section and the results. Below, I provide some specific comments as examples (unfortunately, line numbers are not provided); however, the manuscript needs to be completely revised if you wish to publish it.

  1. Abstract, etc. (entire text) – It is inappropriate to write brackets in italics, while Latin names are incorrectly written without italics! In many places, superscripts for numbers are incorrectly placed.
  2. Keywords – Inconsistent separation, and furthermore, it does not follow the MDPI/Microorganisms template. In fact, the entire manuscript is not written according to the author template.
  3. Chapter 2.2 – Inconsistent labeling of "Gram-...", missing citations with a question mark!! The strain designation should preferably remain as part of the MO name, there is no reason to put it in parentheses – why are you presenting it this way?
  4. Chapter 3.1 – The evaluation of results is absolutely insufficient!!!
  5. Chapter 3.2 – The chapter title is poorly formulated and contains an error in the text (vs.).
  6. Table 1 – The object is inappropriately labeled and needs to be rephrased! This also applies to other objects in the study. It also applies to the text of the chapters – inconsistent labeling of MIC, inappropriate interpretation.
  7. Chapter 3.4.1 – The structure is entirely unsuitable – have you reviewed your own manuscript?? Have you read it??
  8. Discussion – Many errors in citations, many missing citations, inappropriate formulations, inappropriate structure of the text!
  9. Conclusions – The presentation is unsuitable, I recommend clearly writing the sentences and avoiding bullet points.

Author Response

comment 1: Abstract, etc. (entire text) – It is inappropriate to write brackets in italics, while Latin names are incorrectly written without italics! In many places, superscripts for numbers are incorrectly placed.

response 1: Thank you for your comment, we corrected the italics and the supercripts

Comment 2: Keywords – Inconsistent separation, and furthermore, it does not follow the MDPI/Microorganisms template. In fact, the entire manuscript is not written according to the author template.

Response 2: we modified the keywords to follow MDPI/Microorganisms template

Comment 3: Chapter 2.2 – Inconsistent labeling of "Gram-...", missing citations with a question mark!! The strain designation should preferably remain as part of the MO name, there is no reason to put it in parentheses – why are you presenting it this way?

Response 3:thank you for your comment, we modified considering your advice

Comment 4 :Chapter 3.1 – The evaluation of results is absolutely insufficient!!!

Response 4 : thank you for your comment, we reorganized our manuscript and added analysis (complete GC-MS) and precisions about ours results (we added citations to complete ours references)

Comment 5: Chapter 3.2 – The chapter title is poorly formulated and contains an error in the text (vs.).

response 5: thank you for your comment, we modified the title

comment 6: Table 1 – The object is inappropriately labeled and needs to be rephrased! This also applies to other objects in the study. It also applies to the text of the chapters – inconsistent labeling of MIC, inappropriate interpretation.

response 6: We modified the object of Table 1 and we modified and harmonized the units to improve the clarity of the results and the interpretation.

comment 7: Chapter 3.4.1 – The structure is entirely unsuitable – have you reviewed your own manuscript?? Have you read it??

response 7: thank you for your comment, we reviewed entirely the chapter: we added sub-titles and precisions

comment 8: Discussion – Many errors in citations, many missing citations, inappropriate formulations, inappropriate structure of the text!

response 8: thank you for your comment, we added citations to complete ours references ; we reorganized and we added sub-titles.

comment 9: Conclusions – The presentation is unsuitable, I recommend clearly writing the sentences and avoiding bullet points

response 9 : we shortered the conclusion and reorganized the key points to improve the clarity (we suppressed the bullet points as recommanded)

Reviewer 4 Report

Comments and Suggestions for Authors

 The aims of this study were: (i) to evaluate the antibacterial activity of a complex mix of EOs (EO mix) as an MIC based on by macrodilution and microdilution methods for Gram-positive (S. aureus, B. cereus) and Gram-negative bacteria (E. coli, P. aeruginosa), and (ii) to discuss and suggest recommendations for the use of the most appropriate or complementary methods and indicators for MIC determination by the microdilution method. 

The first of all, the introduction discusses antibiotic resistance and the potential use of essential oils in livestock, but it does not clearly connect to the study’s focus on MIC determination methods for EO mixtures against selected bacteria. Consider revising the introduction to emphasize the need for precise MIC evaluation methods and how this study contributes to improving EO-based antibacterial assessments.

Data in Table 1 makes some doubts. In my opinion MIC values are incorrect.
MIC values for essential oils are typically in the range of 0.01–10 µL/mL. But in this study, for example 1.39 × 10³ (1,390 µL/mL) is unrealistically high - far above 100% of the tested volume.

Other comments are marked in the attached file.

Comments on the Quality of English Language

English is good (but I am not a native English speaker).

Author Response

Thank you for your comments; we directly answered on your pdf file. Please, you could find in attach the new version of the manuscript which one with apparent modifications.

Best  regards, 

Round 2

Reviewer 2 Report

Comments and Suggestions for Authors

Dear authors,

your work has certainly improved a lot, but small adjustments could be made (see details inside the document)

Author Response

Please, see the attachment,

Best regards,

Reviewer 3 Report

Comments and Suggestions for Authors

Based on a review of the manuscript, I must state that, compared to the previous version, some problematic passages have been clarified. However, the manuscript still contains many inconsistencies and inadequately described sections. Furthermore, I do not consider the version of the document suitable for the review process—authors should provide the final version of the document with the changes and revisions clearly marked (e.g., text highlighted). The current version is highly confusing and cannot be properly reviewed (the automatic information on the right is unnecessary, and the text mixes the original and current versions, i.e., changes). Specific comments:

  1. To indicate the type of bacteria based on Gram staining, I recommend using "Gram-negative," etc. (with a hyphen! After the last revision, it seems to have been understood differently, but the issue was that the previous version was not consistent in this regard).

  2. Section 3.1 refers to the results of the chemical analysis of EO. However, I do not think such significant results should be presented in just two sentences!! Where is the source for the chemical analysis results (e.g., chromatogram)?

  3. Are you sure the title of section 3.2 is appropriate for publication? In my opinion, it is too complicated, lengthy, and vague.

  4. I doubt that the title of the sections with the results should relate to the method’s name—e.g., section 3.4.2 does not deal with the macro-dilution method itself, but with the results obtained using the macro-dilution method!

  5. Fig. 1, etc. – The microbial counts are typically given to two significant digits, i.e., for example, 3.9x10² instead of 3.91x10². The provided comments are only examples of problematic sections, but it is necessary to go through the entire text and make the necessary revisions throughout the document.

Author Response

  1. To indicate the type of bacteria based on Gram staining, I recommend using "Gram-negative," etc. (with a hyphen! After the last revision, it seems to have been understood differently, but the issue was that the previous version was not consistent in this regard).
    Response 1: thank you for your comment and we modified as your recommandation

  2. Section 3.1 refers to the results of the chemical analysis of EO. However, I do not think such significant results should be presented in just two sentences!! Where is the source for the chemical analysis results (e.g., chromatogram)?

Response 2: we added the table of the 15 compounds of EO mix

3. Are you sure the title of section 3.2 is appropriate for publication? In my opinion, it is too complicated, lengthy, and vague.

response 3: we simplified the title

4. I doubt that the title of the sections with the results should relate to the method’s name—e.g., section 3.4.2 does not deal with the macro-dilution method itself, but with the results obtained using the macro-dilution method!

Response 4: Thank you for your comment, we modified and precized the titles of 3.4.1 and 3.4.2 and 3.4.3 sections

5. Fig. 1, etc. – The microbial counts are typically given to two significant digits, i.e., for example, 3.9x10² instead of 3.91x10². The provided comments are only examples of problematic sections, but it is necessary to go through the entire text and make the necessary revisions throughout the document.

Response 5: thank you for your comment, we modified through the entire manupscript

Reviewer 4 Report

Comments and Suggestions for Authors

I agree with the improvements, just minor corrections must be done in the list of references, i.e. Latin names must be italicized.

Author Response

Thank you for your comment; the list of reference was improved and the Latin names were italicized.